# Infrared-Photoacoustic Spectroscopy and Multiproduct Multivariate Calibration to Estimate the Proportion of Coffee Defects in Roasted Samples

**Rafael Dias [1,\*] , Patrícia Valderrama [2], Paulo Março [2], Maria Scholz [3], Michael Edelmann [1] and Chahan Yeretzian [1]**

1 Coffee Excellence Center, Institute of Chemistry and Biotechnology, Zurich University of Applied Sciences (ZHAW), Einsiedlerstrasse 31, CH-8820 Wädenswil, Switzerland
2 Post-Graduation Program in Food Technology—PPGTA, Federal Technological University of Paraná State—UTFPR, Via Rosalina Maria dos Santos, 1233, Campo Mourão 87301-899, Paraná, Brazil
3 Instituto Agronômico do Paraná—IAPAR, Technical Scientific Board, Rod. Celso Garcia Cid, Km 375, Postal Code 481, Londrina 86001-970, Paraná, Brazil
\* Correspondence: rafael.rafaeltam@gmail.com; Tel.: + 55-47-99696-2300

**Abstract:** Infrared-photoacoustic spectroscopy (IR-PAS) and partial least squares (PLS) were tested as a rapid alternative to conventional methods to evaluate the proportion of coffee defects in roasted and ground coffees. Multiproduct multivariate calibration models were obtained from spectra of healthy beans of *Coffea canephora* and *C. arabica* (Arabica) and blends composed of defective and healthy beans of Arabica in different proportions. The blends, named selections, contained sour, black, broken, whole beans, skin, and coffee woods. Six models were built using roasted and ground coffee samples. The model was optimized through outlier evaluation, and the parameters of merit such as accuracy, sensitivity, limits of detection and quantification, the inverse of analytical sensitivity, linearity, and adjustment were computed. The models presented predictive capacity and high sensitivity in determining defects, all being predicted with suitable correlation coefficients (ranging from 0.7176 to 0.8080) and presenting adequate performance. The parameters of merit displayed promising results, and the prediction models developed for %defects can be safely used as an alternative to the reference method. Furthermore, the new method is fast, efficient, and suitable for in-line application in quality control industrial coffee processing.

**Keywords:** defective coffee beans; coffee quality; roasted coffee; PLS; parameters of merit

## 1. Introduction

Coffee quality is not an easy issue and depends on several factors, mainly biological, geographical, and process criteria such as growing conditions (crop year, altitude, temperature, and other climate particularities), harvesting methods, physical parameters (bean size, color, and shape), composition (species and varieties and occurrence of defects), storage conditions, roasting characteristics, and further processing steps [1]. Among them, the beverage's overall quality, commonly referred to as the 'cup profile' or 'cup quality', is the most relevant factor used globally in coffee trading [2–5]. Since coffee is one of the most important agricultural products in the world and the presence of defective beans in coffee products depreciates the beverage's sensory quality and market value, investigating the occurrence of coffee defects is understandable.

Green coffee defects can be divided into primary and secondary groups. Primary defects include black and sour beans, while the second group is more comprehensive and includes a diversity of terms. The most recurrent are green, partially sour, and partially black beans; broken, damaged by insects, moldy, faded, silver skin, and bored beans; and extraneous matter—e.g., husks, wood, skin, and stones [6]. Obtaining coffee beans from the crop with the desired degree of maturation and a low rate of defects is not an easy task.

Even with modern harvest methods, the proportion of coffee defects in a crop can reach 20% (around 2.6 million bags per year) [7–9], which is a substantial amount. Furthermore, the demanding quality standards of new consumers have been rising increasingly. This class of coffee lovers has sought a more refined product and exquisite sensory properties, even though higher prices are charged for it [10,11]. These specialty coffees are the basis of the "third wave of coffee era" [12].

Coffee processing includes mechanical, manual, and/or electronic sorting to separate and count defective beans. However, after roasting and grinding, recognizing the presence of coffee defects in a sample depends on chemical parameters and sensory assessments. It is noteworthy that the presence of coffee defects in the raw material directly influences the roasted and ground coffee composition and, generally, depreciates the quality of the coffee beverage. For instance, immature beans are associated with bitterness, astringency, and metallic tastes. Sour beans contribute to an overfermentation flavor caused by humidity and temperature adversities during storage, transportation, and processing. Black beans are usually experienced as having an ashy and heavy flavor. Black beans can be overripe cherries, dead beans within the cherry, or beans damaged by pests [13]. Broken beans, wood, stones, husks, twigs, and other fruit tissues also influence the flavor of a roasted coffee due to its composition (different from that of whole and healthy beans) or by interfering with the roasting process. It should be noted that even trained and certified tasters cannot guarantee sufficient precision and sensibility in sensory determinations for identifying and quantifying coffee defects [14].

Specific groups of compounds (e.g., chlorogenic acids, amino acids, diterpenes, carbohydrates, tocopherols, triglycerides, volatile metabolites, and other compounds) or key substances (e.g., caffeine, 16-*O*-methyl-cafestol, trigonelline, and nicotinic acid) in roasted coffee has been determined by classical techniques, such as chromatography-based methods (high-performance liquid chromatography [15–17], gas chromatography [18–21]), mass spectrometry [22], and UV-Vis spectroscopy after colorimetric reaction [23]. The contents of these compounds are used as sample discriminating parameters since coffees of varied quality and/or blended with different species and varieties may present different compositions.

Considering a modern analytical approach, coffees with varying quality are evaluated by direct, faster, and eco-friendly analytical methods, mainly including variations of spectroscopic approaches, such as near-infrared (NIR) [5,24–26], mid-infrared [27,28], Vis-NIR [29], and Raman [14,30] spectroscopies.

Infrared spectroscopy is an analytical technique based on the bond vibrations of atoms in a molecule. The spectrum originates from the absorption of a particular fraction of energy from the incidence of IR radiation. The signal generated is registered by a detector [31]. A common detector of signal analysis is based on the reflectance of radiation. A diversity of commercial reflectance sampling accessories is accessible to suit a variety of sample types [32]. While the mid-infrared region (4000–400 cm$^{-1}$) produces signals from fundamental vibrations of bonds related to the compounds present in the sample, the NIR spectrum (800–2500 nm) provides information regarding the molecular absorptions of overtones and combinations of fundamental vibration bands in the mid-IR region [33]. The principle of the study that uses NIR spectra as a fingerprint for coffee samples lies precisely in the differences in the spectra generated by variations in the composition of the samples. Since the presence of coffee defects influences the final composition of the roasted and ground product, variations in the spectra will also occur. A suitable way to perceive these variations is through chemometric tools. Craig et al. (2014) [27] introduced a comparative analysis of the performances of FTIR (Fourier transform infrared) and NIR for the qualitative differentiation of roasted non-defective and defective coffees. The elastic net models presented a high level of classification. The correct classification of non-defective coffees was attributed to absorbance regions that are characteristic of lipids (1722–1759 cm$^{-1}$, 2810–2848 cm$^{-1}$, 2908–2920 cm$^{-1}$, 1680–1755 nm, and 2132–2166 nm) and carbohydrates (1138–1165 cm$^{-1}$ and 1760–1871 nm).

Recently, Araújo et al. (2021) [34] used NIR spectroscopy and digital images (from chemometrics-assisted color histogram-based analytical systems—CACHAS) for the non-destructive authentication of roasted and ground coffees with different quality denominations (gourmet, traditional, and superior). One-class partial least squares and data-driven soft independent modeling of class analogy (DD-SIMCA) were applied as one-class classifiers and evaluated in terms of specificity and sensitivity. DD-SIMCA was applied using an offset correction for NIR and an RGB histogram for CACHAS, recognizing all the 90 samples in both the training and test sets. In conclusion, the authors highlighted that NIR spectra allow a definitive authentication due to the molecular character of the analytical information contained in the relation between the sample composition and spectral data.

Proposing a real-time analysis for the industry quality control of coffee, Baqueta et al. (2021) [5] used a handheld near-infrared spectrometer combined with partial least squares with discriminant analysis (PLS-DA) to directly assess cup profiles in roasted and ground coffee blends. The model presented sensitivity and specificity from 91 to 100%, 84–100%, and 73–95% in the training, prediction, and internal cross-validation sets, respectively. With the encouraging results presented, the new method will help coffee professionals in their decisions during the evaluation of the cup in subsequent tests on an industrial scale. In another report, NIR spectroscopy coupled with PLS was successfully used to assess the color, granulometry, moisture percentage, and infusion time of roasted coffees [25].

In the meantime, it is worth highlighting the use of photoacoustic spectroscopy for food analysis [35]. When intensity-modulated radiation is absorbed by a sample isolated in an acoustic chamber and filled with inert gas, the photoacoustic effect is observed. The sample generates heat due to a reabsorption process, and the absorbed energy is released as heat, which causes temperature oscillation, generating periodic acoustic waves. A very sensitive microphone detects the resulting pressure changes and converts them into an electrical signal. Fourier transformation (FT) of the resulting signal generates a characteristic infrared spectrum (the technique is abbreviated as FTIR-PAS) [31]. Thus, it combines the utility of interferometry with the standard sample-gas microphone of the photothermal technique for a depth-profile analysis of samples. Although the absorption spectrum is retrieved from IR-PAS experiments, the thermal behavior of the sample rather than the optical properties plays a major role in the generation of the PA signal [35].

The applications of photoacoustic technique range from investigations on solids and liquids to gases and life sciences. Agricultural and environmental sciences [36,37], processes in surfaces and thin films and biofilms [38,39], air quality monitoring [40], and pollutants in liquids [41] are some of the topics explored by the technique. In recent years, few papers have been published using PAS spectroscopy for food analysis. FTIR-PAS was used for the determination of total protein and wet gluten in wheat flour. The evaluation of the prediction performance for both analytes was carried out by multivariate calibration models. Root mean square errors of prediction (RMSEPs) of 0.362% and 0.229%, with coefficients of determination of 0.84 and 0.96 for total protein and wet gluten were obtained, respectively. External validation produced coefficients of determination higher than 0.82 for total protein and wet gluten models. The results presented a viable FTIR-PAS spectral dataset that can predict the protein level in wheat flour [42]. Photoacoustic spectroscopy was applied to assess the impact of moringa at different levels on the elaboration, texture, sanity, and color of wheat bread. The plant moringa oleifera has been used for centuries for its medicinal properties and health benefits; it presents a high content of antioxidant and oily substance that nourishes the human skin. The most significant statistical differences were observed in the spectrum region of 300–450 nm when comparing the control bread and the moringa-added samples. Interesting results were verified. Among them, it is highlighted that the photoacoustic signal amplitude of bread increases with the moringa percentage, and a positive correlation was observed between the photoacoustic signal and the number of fungal colonies in bread. The study showed that PAS spectroscopy can be used to evaluate the quality of bread in different formulations [43].

Reports using infrared-photoacoustic spectroscopy to investigate roasted and ground coffee are still scarce in the literature. FTIR-PAS assessed adulterated coffee samples with barley, corn, and coffee parchment with satisfactory discrimination results. Even though the work was published almost three decades ago, the concern of coffee researchers and regulatory agencies regarding the adulteration of roasted and ground coffee is notable. The authors considered that adulteration was commonly accomplished by the addition of roasted and ground cereal grains, or their equivalent, which was a relevant problem for those engaged in quality control in the packaged coffee industry [44]. Another study reported the possibility of discriminating organic from non-organic coffees by comparing the respective PAS spectra [45]. Recently, this research group developed a method based on infrared-photoacoustic spectroscopy measurements and chemometrics to differentiate coffees containing genuine defects occurring in coffee crops [8,46]. PLS-DA allowed the prediction of the amount and type of specific defects in blends, while a principal component analysis revealed similarities between them. A successful predictive model was built using six classes of blends. The model could classify all samples into four classes.

The main goal of the present study was to use Brazilian coffees to build and validate multivariate calibration models for quantifying the percentage of sour, black, broken, whole beans, skin, and wood using the FTIR-PAS coupled with multivariate calibration by partial least squares. For this purpose, accuracy, sensitivity, analytical sensitivity, linearity, adjustment, and limits of detection and quantification were estimated as parameters of merit. The model results were compared with reference values to confirm the applicability of the proposed method.

## 2. Materials and Methods

The experimental measurements and data acquisition were conducted at Zurich University of Applied Sciences, in the Wädenswil campus, Switzerland. The statistical experiment and data evaluation were conducted at the Federal Technological University of Paraná, Campo Mourão campus, Brazil.

### 2.1. Coffee Samples

Instituto Agronômico do Paraná (IAPAR, Londrina, Paraná, Brazil; climate conditions: humid subtropical, Latitude$-$23.29, Longitude$-$51.17; 23°17′34″ S, 51°10′24″ W) supplied the samples of coffee. Healthy beans (whole beans without defects) of *Coffea canephora* (Robusta) and *C. arabica* (Arabica) (named basis) and 25 blends (named selections) composed of defective coffee and healthy Arabica beans were used in different proportions to build the calibration models. The selections contained sour, black, broken, whole beans, skin, and coffee woods. The samples (the bases, the selections, and the blends) are described in detail by Dias et al. (2018) [46]. Bean by bean, professional specialists in coffee quality from Instituto Agronômico do Paraná (Brazil) hand-picked the whole and healthy beans (sour, black, and broken beans), wood, and skin of each selection. The visual and tactile characteristics were considered. The counting was performed based on the weight expressed as a percentage.

The final blends comprised one selection (20 or 40%) and one basis (80 or 60%). Three such bases were used: 100% Arabica coffee and two blends of Arabica to Robusta in the proportions of 80:20 and 50:50 (*w/w*). Thus, each selection was blended with the three types of bases in two ratios, generating 150 samples plus 4 different samples: the three bases and 100% Robusta coffee. Considering the triplicate analysis, 462 spectra were recorded and evaluated.

Coffee samples were roasted (Probat Emmerich am Rhein, Germany, model PRG1Z, ERD Gas) to a medium degree (17% weight loss and 22 to 26 of luminosity, L* $-$ Konica Minolta portable colorimeter BC-10). The roasted coffee samples were ground (setting 2) on a Ditting grinder/KR805 (Bachenbülach, Switzerland).

### 2.2. FTIR-PAS Spectroscopy Assays

The samples of roasted coffee blends were submitted to FTIR-PAS spectroscopy analysis (16 scans; 4 cm$^{-1}$ resolution; 600–4000 cm$^{-1}$). An FTIR spectrometer (Bruker/Billerica, MA, USA), model Tensor 37 coupled to a Gasera photoacoustic detector model PA 301 (Turku, Finland), interfaced with a DSP Module was used. The experimental conditions were previously set up [8]. The normalized signal generated the PAS spectrum, whose profile depended on the sample's composition. FTIR-PAS spectra were used for the multivariate model development. The instrumental noise was estimated by recording FTIR-PAS spectra without samples.

### 2.3. Multivariate Calibration

Partial least squares (PLS) have been discussed in detail in proper reports [47–49]. PLS is a regression tool applied to first-order data for multivariate calibration. Here, we consider that the method is also multicomponent due to the different species used in the sample blends. Some advantages are more relevant to a multiproduct model, e.g., a restricted number of computations are necessary for each prediction in routine analysis, saves time on updates, and produces robust models [47].

The multivariate multicomponent calibration computes latent variables (LVs) by targeting the highest possible covariance between **X** and **y** [48,49]. In this case, the data matrix **X** was constituted by the FTIR-PAS spectra of the coffee blends, and the vector y contained the reference values, i.e., the proportion of the different defects of coffee.

Data preprocessing and chemometric models were performed in Matlab (software version R2007b) (The MathWorks Inc., Natick, MA, USA) and PLS-Toolbox computational package version 5.2 (Eigenvector Technologies, Manson, WA, USA). The spectra had the baseline corrected through the baseline algorithm from PLS Toolbox, and they were smoothed by the Savitzkyl-Golay (savgol) algorithm (5 points window and first-order polynomial) [50]. PLS regression models were built with mean center data preprocessing.

In the calibration step 100 samples were used, while in the external validation step 54 samples were used; all were pre-selected by the Kennard-Stone algorithm [51]. The choice of the number of latent variables (LVs) was based on the lowest result for the root mean square of the cross-validation error (RMSECV) [48] with contiguous blocks of 10 samples. In addition, the percentage of explained variance in the **y** block was also considered for choosing the number of LVs [26].

Outliers were evaluated to improve model accuracy, which is a result of eliminating samples with extreme leverage (as they exhibit high influence on the model) and eradicating unmodeled residuals in **X** and **y** (the strategy is based on recommendations from the Standard Practices for Analysis Quantitative Multivariate by Infrared—ASTM) (E1655-00) [52].The multivariate analytical validation of PLS models was based on determining parameters of merit following previous multivariate calibration studies [24–26,53]. These parameters included accuracy, adjust, linearity, analytical sensitivity, and limits of detection and quantification.

The regression coefficient vector was considered for each PLS model to find which spectral regions were important to model and predict the proportion of defects in roast and ground coffee.

## 3. Results and Discussions

The FTIR-PAS spectra (600–4000 cm$^{-1}$) of 154 samples of ground and roasted coffee blends with different proportions of defects and species were obtained and displayed, after preprocessing, as shown in Figure 1.

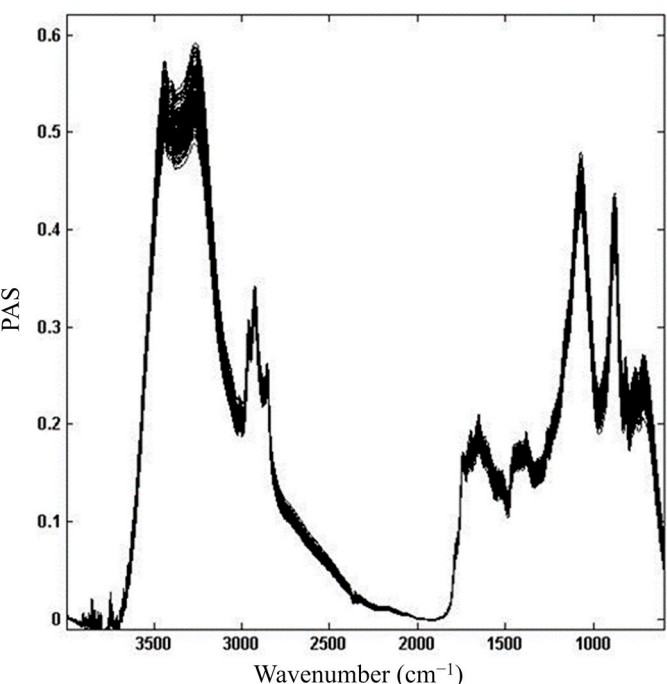

**Figure 1.** FTIR-PAS spectra of coffees after preprocessing.

Undesired random or systematic variations can be minimized by mathematical corrections named preprocess, a preliminary step when applying chemometrics. The analytical signals are rarely submitted in their raw form to chemometric modeling [54]. Among the most common systematic signal variations, baseline shifts (offsets) and noise are very common and can be addressed by a given mathematical transformation. The smoothing is applied to reduce the aleatory component of a data set and increase the signal-to-noise ratio. Here the smoothing was done through a savgol algorithm. An instrumental or sampling problem can shift the spectra on the ordinate axis [55]. The shift is not related to the sample content and can compromise the result of a chemometric approach.

The bands between 3000 and 3600 cm$^{-1}$ influenced the blends differentiation, while bands located in 1067 cm$^{-1}$ (pyruvic acid, pyridine, and quinic acid), 3356 cm$^{-1}$ (chlorogenic acids), and between 1000 and 1750 cm$^{-1}$ (trigonelline and caffeine alkaloids) were correlated to specific compounds [8].

PLS models were constructed for each considered defect (PLS1 models) by using 8 LVs for sour, skin, and broken; 9 LVs for black and whole; and 11 LVs for woods (Table 1). The number of LVs were chosen considering RMSECV and the percentage of variance explained in the **y** block.

The outlier samples were evaluated to optimize the PLS models. A total of 21, 17, 22, 15, and 16 outliers were identified for the models in the sour, for both black and skin, whole, broken, and wood, respectively. According to the ASTM (E1655-17) [56] and previous works [25,57], the outliers can be removed from up to 22.2% of the total number of samples in the dataset. In the present research, all the models agree with these recommendations.

Considering a reliable application for routine analysis, adequate validation is mandatory to certify the predictive capacity of the model, which is based on the determination of parameters of merit [39]. Parameters of merit for first-order multivariate calibration were described in previous reports [53,58–63]. The results for the parameters of merit obtained from PLS models (after outliers' elimination) are presented in Table 1.

Root mean square error of calibration (RMSEC) and prediction (RMSEP) represent the accuracy for the PLS models (Table 1). These parameters report the closeness of agreement between the reference value and the value found by the calibration model. For

the modeled parameters, sour achieved higher accuracy, which is directly related to the modeled reference values for that parameter, being larger than the others.

Furthermore, regarding the RMSEC and RMSEP results, it is possible to affirm that the models' dimensions were properly chosen, and the models were not overfitted since the RMSEC and RMSEP results for all models were close [64]. The number of LVs affects the parameters of merit, especially accuracy and, consequently, the others. Moreover, depending on the application of a suitable preprocess, the correct number of LVs allows for the RMSEC and RMSEP closeness results, i.e., the most appropriate model.

Adjustments can also certify the accuracy by plotting the percentage of the specific defects (%D) determined by the reference method (selection and counting of the different defects made by a trained professional) against %D determined by the PLS model (Figure 2). The coefficient (R-squared) is presented in Table 1. The model for the sour defect presented the highest correlation coefficients, around 0.8, whereas for the other defects, the correlation coefficients are in the range of 0.71 to 0.78. A correlation coefficient of up to 0.7 can be considered satisfactory for multivariate calibration models elaborated from results obtained with high variability (as in this case) [26].

**Table 1.** Analytical parameters of merit for the multiproduct PLS model.

| Parameters of Merit | | Equation | Sour (8 LVs [a]) | Black (9 LVs) | Skin (8 LVs) | Whole (9 LVs) | Broken (8 LVs) | Woods (11 LVs) |
|---|---|---|---|---|---|---|---|---|
| Modeled range | | - | 0–29.49% | 0–17.06% | 0–6.28% | 0–11.43% | 0–11.27% | 0–1.02% |
| Accuracy (%) | RMSEC [b] | $=\sqrt{\sum_{i=1}^{nc}\frac{(y_i-\hat{y}_i)^2}{nc-nVL+1}}$ | 2.8074 | 1.9244 | 0.7642 | 1.2318 | 1.4005 | 0.0609 |
| | RMSEP [c] | $=\sqrt{\sum_{i=1}^{nv}\frac{(y_i-\hat{y}_i)^2}{nv}}$ | 2.7979 | 1.6406 | 0.7364 | 1.2471 | 1.3855 | 0.0574 |
| Sensitivity (%) [d] | | $=\frac{1}{\|b\|}$ | 0.0022 | 0.0021 | 0.0079 | 0.0034 | 0.0053 | 0.0503 |
| Analytical sensitivity$^{-1}$ (%) [e] | | $=\left(\frac{Sensitivity}{\|\delta x\|}\right)^{-1}$ | 0.4072 | 0.3563 | 0.1170 | 0.2240 | 0.1748 | 0.0128 |
| Limit of detection (%) [e] | | $=\frac{3.3\delta x}{Sensitivity}$ | 1.3438 | 1.1756 | 0.3862 | 0.7394 | 0.5769 | 0.0421 |
| Limit of quantification (%) [e] | | $=\frac{10\delta x}{Sensitivity}$ | 4.0721 | 3.5626 | 1.1702 | 2.2405 | 1.7482 | 0.1276 |
| Fit (R-squared) | | | 0.8080 | 0.7416 | 0.7675 | 0.7863 | 0.7176 | 0.7430 |
| Linearity | Jarque-Bera test [f,g] | | A = 1.874 B = 4.774 | A = 0.866 B = 4.8467 | A = 0.955 B = 4.847 | A = 1.420 B = 4.692 | A = 2.114 B = 4.869 | A = 0.896 B = 4.799 |

[a] LVs = Latent Variables; [b] RMSEC: root mean square error of calibration, where *nc* is the number of the samples in the calibration set, $y_i$ is the reference value of the sample *i*, and $\hat{y}$ is the predicted value of the sample *i* ("+1" is added when the data are mean center); [c] RMSEP: root mean square error of prediction, where *nv* is the number of samples in the validation set, $y_i$ is the reference value of the sample *i*, and $\hat{y}$ is the predicted value of the sample *i*. [d] b = regression coefficients vector; [e] $\delta x$ = estimate for the instrumental noise. Parameters of the Jarque-Bera test: [f] A = JBSTAT, [g] B = CRITVAL.

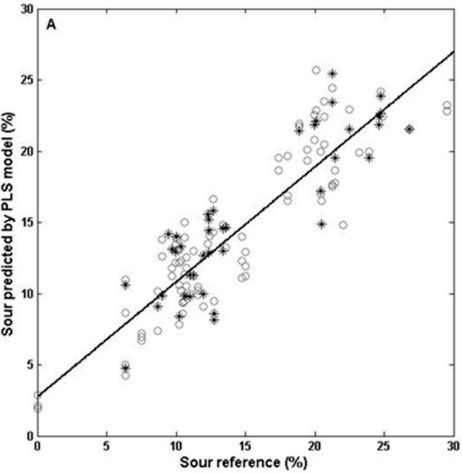
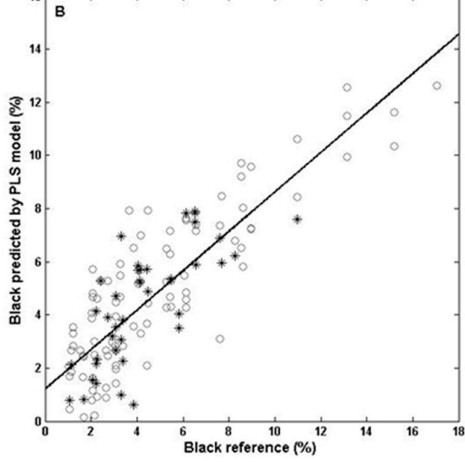

**Figure 2.** *Cont.*

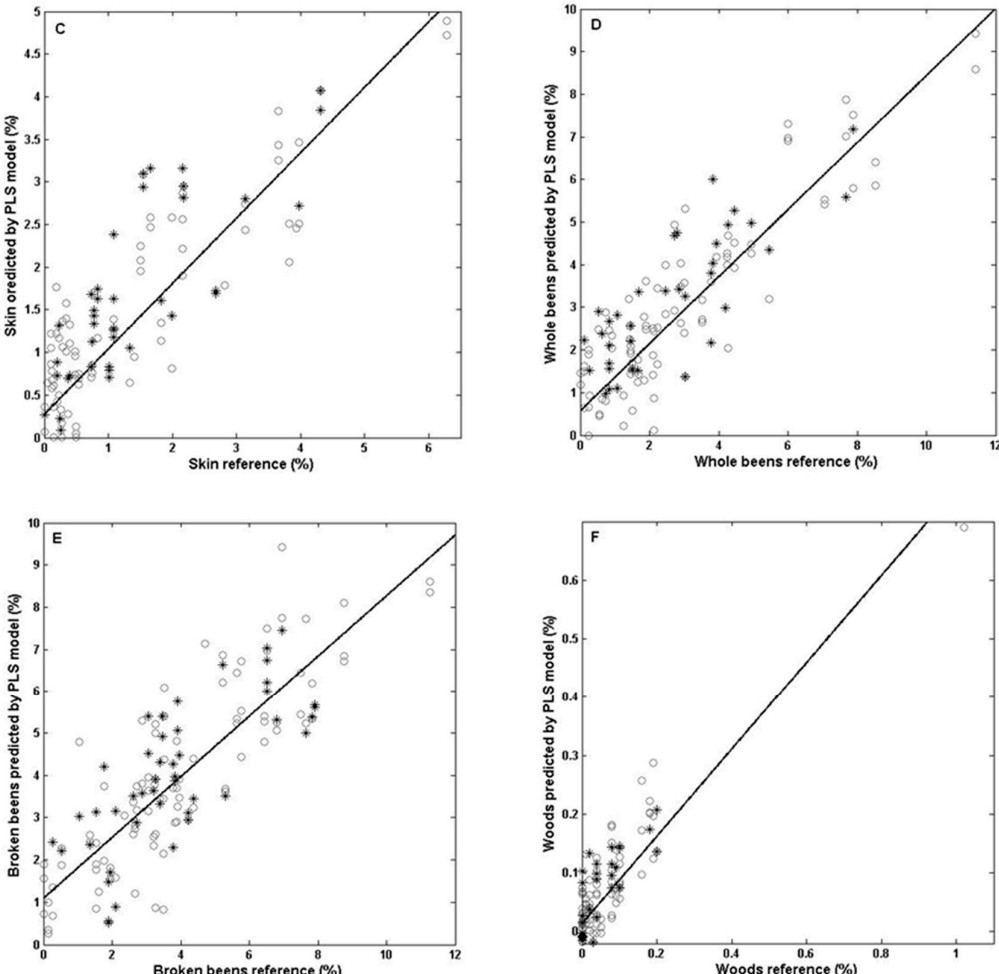

**Figure 2.** Adjustment (after the elimination of outliers). Reference values against values predicted by the PLS models: (**A**) sour; (**B**) black; (**C**) skin; (**D**) whole; (**E**) broken; and (**F**) woods. (o) Calibration samples and (∗) validation samples.

Sensitivity corresponds to the fraction of the analytical signal due to increasing the level of a particular analyte by a unit concentration. Considering inverse multivariate calibration models, such as PLS, it is usually determined by the inverse of the regression coefficients vector. The sensitivity parameter generates complex results that are difficult to judge due to the preprocessing used in the PLS. On the other hand, analytical sensitivity is more elementary and informative for comparing and evaluating the sensitivity of an inverse multivariate calibration method [64]. The analytical sensitivity$^{-1}$ allows the establishment of a minimum percentage of defects that is discernible by the PLS model. Thus, it is possible to discriminate samples with defects of 0.4072% for sour, 0.3563% for black, 0.1170% for skin, 0.2240% for whole, 0.1748% for broken, and 0.0128% for woods.

The limit of detection shows the lowest percentage of defects that can be detected but not necessarily accurately quantified. The limit of quantification is the lowest percentage of defects that can be quantified with accuracy. Limits of detection and quantification for the models presented coherent results with the measured quantities and considering the modeled range. Then, the quantification limits obtained showed that samples with sour, black, skin, whole, broken, and woods lower than 4.0721%, 3.5626%, 1.1702%, 2.2405%, 1.7482%, and 0.1276%, respectively, are not quantified with accuracy.

Residual plots from calibration and validation samples (Figure 3) were used to check the linearity of the PLS model. The residual distribution has a random behavior, which reinforces that the data fit into a linear model. To confirm the random distribution of residuals, an appropriate statistical test (Jarque-Bera test [65]; 95% confidence) was per-

formed. This is a goodness-of-fit test of departure from normality, based on the sample skewness and kurtosis. The results (Table 1) indicated the randomness of the residuals. When the Jarque-Bera test (JBSTAT) < CRITVAL, the null hypothesis (residuals are normally distributed with unspecified mean and standard deviation) can be accepted at a significance level of 95%.

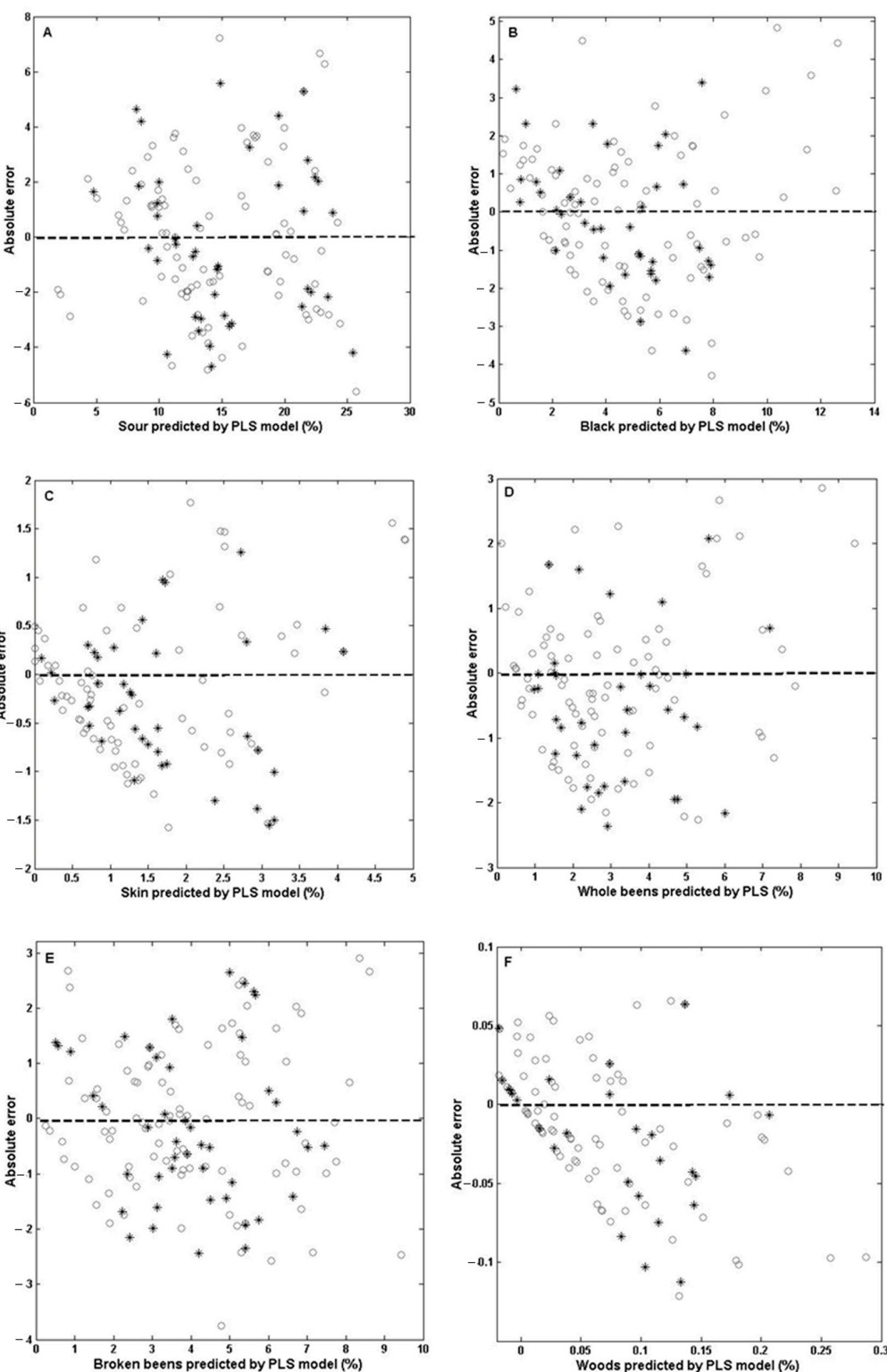

**Figure 3.** Residuals: (**A**) sour; (**B**) black; (**C**) skin; (**D**) whole; (**E**) broken; (**F**) woods. (o) Calibration samples and (∗) validation samples.

The new analytical method developed based on IR-PAS spectroscopy and multiproduct and multivariate calibration is promising. The concept of assessment of different species of healthy coffee and coffee defects in blends in the same model reinforces the heterogeneity of samples, which makes the model more comprehensive, reliable, and realistic. The method merits attention both for its analytical efficiency and for its practicality, cleanliness, and fastness, and does not require large investments.

## 4. Conclusions

A new method for coffee analysis was developed based on IR-PAS and chemometric regression PLS. The percentage of defects from roasted and ground Brazilian coffee was predicted. The models were validated by the parameters of merit, evidencing that the PLS models are sensitive to discriminate, detect, quantify, and predict the percentage of defects sour, black, skin, broken, wood, and whole beans on the coffee samples. When compared to the selection and counting of the different defects made by a trained professional, the proposed method is a promising alternative for the coffee industry to evaluate coffee defects without the high dependence on the analyst's perception.

**Author Contributions:** Conceptualization, R.D., M.S. and C.Y.; methodology, R.D. and M.E.; data analysis, P.M. and P.V.; investigation, R.D.; resources, M.S. and C.Y.; writing—original draft preparation, R.D.; writing—review and editing, P.V. and C.Y.; supervision, C.Y.; project administration, R.D. and C.Y. All authors have read and agreed to the published version of the manuscript.

**Funding:** The research was funded by the Brazilian National Council for Scientific and Technological Development (CNPq) [process 202515/2014-1] and [process 306606/2020-8].

**Data Availability Statement:** Data are available from the authors upon request.

**Acknowledgments:** The authors are grateful to Zurich University of Applied Science; Science without Borders, a student program of the Brazilian Government; the Federal Technological University of Paraná State; and Agronomic Institute of Paraná.

**Conflicts of Interest:** There are no conflict of interest according to the authors.

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
