# Peer review of "Infrared-Photoacoustic Spectroscopy and Multiproduct Multivariate Calibration to Estimate the Proportion of Coffee Defects in Roasted Samples"

_beverages, doi:10.3390/beverages9010021_

Round 1

Reviewer 1 Report

The author developed a multivariant model based on infrared-photoacoustic spectroscopy and partial least squares to differentiate blends of healthy and defective beans of Coffea canephora and C. arabica. Several kinds of defects were analyzed in the PLS analysis including sour, black, broken, whole beans, skin, and coffee woods. The effectiveness of the model was evaluated based on accuracy, sensitivity, the inverse of analytical sensitivity, limits of detection and quantification, adjust, and linearity. The PLS showed linearity between the prediction and reference, which suggests the reliability of the model. The author further demonstrated that the residuals were randomly distributed indicating fitting to a linear model. This work presented a thorough analysis of the coffee defects. Before accepting it for publication, the following questions would need to be addressed.

1.       If my understanding was correct, the accuracy is defined as the error of the validation, which means a higher ‘accuracy’ represents a larger deviation of the predicted value to the reference value. Do the authors have any hypothesis that accounts for the higher ‘accuracy’ for ‘sour’?

2.       In terms of the preprocessing of the PAS spectra, do the baseline correction and spectral smoothing affect the validation results? Can the author show some comparisons before and after preprocessing?

3.       How would the choice of the number of latent variables affect the figure of merits? Can the author elaborate a bit more on this point?

4.       Figure 2 was not directly mentioned in the main text. It would be helpful if the author could add a few sentences to address figure 2.

5.       The unit of g kg-1 for ‘wood’ is missing in line 210.

Author Response

First, we would like to thank each reviewer for their kind feedback. All notes were considered valuable for the quality of the manuscript. We consider all comments and recommendations from the Editor and Reviewers. Below you can find the comments on each revised point.

1.       If my understanding was correct, the accuracy is defined as the error of the validation, which means a higher ‘accuracy’ represents a larger deviation of the predicted value to the reference value. Do the authors have any hypothesis that accounts for the higher ‘accuracy’ for ‘sour’?

Response: Yes, you correct understanding. The higher accuracy for the sour is related to the modeled range. In fact, the accuracy is directly related with the modeled reference values that for sour is large than for the other parameters. A line with modeled range was included in Table 1, and this explanation was included in the manuscript (Paragraph of line 268). We are sorry for this failure.

2. In terms of the preprocessing of the PAS spectra, do the baseline correction and spectral smoothing affect the validation results? Can the author show some comparisons before and after preprocessing?

Response: The baseline correction and smoothing positively affect the validation results. We include a discussion in the manuscript to highlight the importance of chosen a correct preprocessing (Paragraph of line 237).

We performed a PLS model with the raw data for the parameter sour (that present higher accuracy among the modeled parameters). All the models were performed with mean center and 8LVs. The difference between the RMSEC and RMSEP results by using raw spectra confirm that the error behavior is not the same in the calibration and validation sets. On the other hand, the closeness results with preprocessed spectra shows that the error came from the same source.

Accuracy

preprocessed / raw

(Before outlier evaluation)

preprocessed / raw

(After outlier evaluation)

RMSEC

3.3920 / 3.3021

2.8074 / 2.9792

RMSEP

4.8637 / 15.9625

2.7979 / 7.6486

  1. How would the choice of the number of latent variables affect the figure of merits? Can the author elaborate a bit more on this point?

Response: The number of LVs affect the figures of merit specialty concerning accuracy, and by consequence the other ones. When a correct preprocess was applied, the correct number of LVs bring RMSEC and RMSEP closeness results. This discussion was included in the manuscript (Paragraph of line 275).

  1. Figure 2 was not directly mentioned in the main text. It would be helpful if the author could add a few sentences to address figure 2.

Response: Figure 2 was directly mentioned now.

  1. The unit of g kg-1 for ‘wood’ is missing in line 210.

Response: It was adjusted. Thank you!

Reviewer 2 Report

Manuscript entitled “Infrared-photoacoustic spectroscopy and multiproduct-multivariate calibration to estimate the coffee defects proportion in roasted samples” reports a simple application applying infrared spectroscopy based PLS models to predict the proportion of six types of defects in samples of roasted ground coffee from two species. Six types of defects were studied.

The manuscript is well written. Results are objective and clear. Some specific comments follow:

1- The characteristics of each type of defect can affect the IR spectral profile and the performance of the models in a particular way. For example, some defects may be mostly related to the color of the grain, others to the composition, shape, texture, and so on. Therefore, it is information relevant to the proposed application and I suggest including a discussion on it in the introduction.

2- Please, provide a description for the reference method employed to determine the proportion of defects in each sample.

3- In lines 150-152, the authors states that “…the regression coefficient vector was considered for each PLS model seeking to find which spectral regions were important for modeling and predicting…”

The regression coefficients were not shown in the manuscript and I did not find any discussion on the influence of the spectral regions for the predictive ability of the PLS models.

4- Please indicate if spectra in Figure 1 are raw or corrected.

5- Please adopt one of the two terms “parameter of merit” or “figure of merit” to use in the manuscript.

6- As indicated in Table 1, please provide a description of the method used to obtain an estimate of the instrumental noise employed in the calculation of analytical sensitivity, limit of detection and limit of quantification.

- Also, Table 1 index order needs to be revised.

7- In lines 206-210: “Limits of detection and quantification for the models presented coherent results with the measured quantities and since their ranges were between 6.33 and 29.5 g kg-1 (sour), 1.03 and 17.2 g kg-1 (black), 0.12 and 19.6 g kg-1 (skin), 60.2 and 91.4 g kg-1 (whole/healthy), 0.13 and 15.0 g kg-1, and 0.01 and 2.04 (woods).”

It seemed out of place in the text. Quantification limits are presented in this sentence and again in the following sentence with different values. Are the reference values in units of g/kg? It had not been mentioned up to this point. Please review this point and make it clearer.

8- Please clarify if the models described in Table 1 and Figure 2 are before or after outliers removal. For example, in Figure 2f, the sample with reference value for woods beyond 1% should be removed.

As in the case with models for black, skin and whole defects, I suggest the external validation samples to be more evenly distributed, especially in the higher percentage region.

9- Finally, I suggest a spell check for English.

Author Response

First, we would like to thank each reviewer for their kind feedback. All notes were considered valuable for the quality of the manuscript. We consider all comments and recommendations from the Editor and Reviewers. Below you can find the comments on each revised point.

1- The characteristics of each type of defect can affect the IR spectral profile and the performance of the models in a particular way. For example, some defects may be mostly related to the color of the grain, others to the composition, shape, texture, and so on. Therefore, it is information relevant to the proposed application and I suggest including a discussion on it in the introduction.

Response: Part of the Introduction has been reworked to meet this and other requests from the reviewers. New information was added to make it clear that there are different defects in coffee, and these influence the profile of the spectrum obtained via NIR, especially because of the variations in its chemical composition.

2- Please, provide a description for the reference method employed to determine the proportion of defects in each sample.

Response: Information was added regarding the method of selection and counting of defects (Paragraph of line 167). It is worth remembering that, as already mentioned in the line167, there is a published paper that shows in detail how these samples were obtained by our research group (Dias, R.C.E.; Valderrama, P.; Marco, P.H.; Dos Santos Scholz, M.B.; Edelmann, M.; Yeretzian, C. Data on roasted coffee with specific defects analyzed by infrared-photoacoustic spectroscopy and chemometrics. Data in Brief 2018, 20, 242–249)

3- In lines 150-152, the authors states that “…the regression coefficient vector was considered for each PLS model seeking to find which spectral regions were important for modeling and predicting…” The regression coefficients were not shown in the manuscript and I did not find any discussion on the influence of the spectral regions for the predictive ability of the PLS models.

Response: We are sorry. This was a real mistake. It was removed.

4- Please indicate if spectra in Figure 1 are raw or corrected.

Response: We agree. The corrected spectra are present. The correction was done in the Figure caption and in the manuscript.

5- Please adopt one of the two terms “parameter of merit” or “figure of merit” to use in the manuscript.

Response: We agree. We correct all for parameter of merit.

6- As indicated in Table 1, please provide a description of the method used to obtain an estimate of the instrumental noise employed in the calculation of analytical sensitivity, limit of detection and limit of quantification.

Response: We agree. We included a description in the section 2.2 “The instrumental noise was estimated by recording FTIR-PAS spectra without samples.”

- Also, Table 1 index order needs to be revised.

Response: It was revised.

7- In lines 206-210: “Limits of detection and quantification for the models presented coherent results with the measured quantities and since their ranges were between 6.33 and 29.5 g kg-1 (sour), 1.03 and 17.2 g kg-1 (black), 0.12 and 19.6 g kg-1 (skin), 60.2 and 91.4 g kg-1 (whole/healthy), 0.13 and 15.0 g kg-1, and 0.01 and 2.04 (woods).” It seemed out of place in the text. Quantification limits are presented in this sentence and again in the following sentence with different values. Are the reference values in units of g/kg? It had not been mentioned up to this point. Please review this point and make it clearer.

Response: We agree. We are sorry for this mistake. We correct this point in the manuscript.

8- Please clarify if the models described in Table 1 and Figure 2 are before or after outliers removal. For example, in Figure 2f, the sample with reference value for woods beyond 1% should be removed.

Response: We agree. We include that the results presented in Table 1 and Figure 2 are both after outliers elimination. We prefer keep the sample around 1% because the modeled range for woods are from zero to 1.02%. By removing the samples around 1% all parameters of merit for this parameter will be change.

As in the case with models for black, skin and whole defects, I suggest the external validation samples to be more evenly distributed, especially in the higher percentage region.

Response: In all cases the samples for calibration set and external validation set were chosen based on Kennard-Stone algorithm. This information was clarified in the manuscript (line 216).

9- Finally, I suggest a spell check for English.

Response: It was done. Thank you!

Reviewer 3 Report

The research paper is interesting.

But I think there are a lot of similar papers to yours.

I do not see anything new that is offered by the authors.

Good luck.

Author Response

The research paper is interesting.

But I think there are a lot of similar papers to yours.

I do not see anything new that is offered by the authors.

Good luck.

Response: Thank you!

Reviewer 4 Report

Dear Authors

The study deals with a meaningful and important question, so the topic is interesting to certain readers in Beverages. Also, the chosen approach is comprehensible and was carried out correctly. However, the following few comments should be answered before a final recommendation.

1.      In the introduction section, please briefly explain IR-PAS, such as the advantages of this technique, whether any other same study has been done using this technique with other samples, etc, because it may be new for some readers.

2.      As we know, chemical composition also depends on geographical origin (climate, locations, etc). In your paper, because you only used coffee from Brazil so your model will be restricted in the detection of coffee defects from Brazil, not from all over the world. So, I think in your title, introduction, results and discussion, you should have mentioned this limitation by saying that the detection method is for the identification of coffee defects from Brazil

3.      Preprocessing spectra is one of the important steps when you would like to build a prediction model. Please give an explanation in the methodology of what kind of normalization method you used or any other preprocessing method you have applied in your spectra before analyzing with PLS

Author Response

First, we would like to thank each reviewer for their kind feedback. All notes were considered valuable for the quality of the manuscript. We consider all comments and recommendations from the Editor and Reviewers. Below you can find the comments on each revised point.

  1. In the introduction section, please briefly explain IR-PAS, such as the advantages of this technique, whether any other same study has been done using this technique with other samples, etc, because it may be new for some readers.

Response: Interesting point. We have added information on IR-PAS, advantages, and references regarding the use of this technique in coffee analysis (Paragraphs of lines 131 and 144). Thank you.

  1. As we know, chemical composition also depends on geographical origin (climate, locations, etc). In your paper, because you only used coffee from Brazil so your model will be restricted in the detection of coffee defects from Brazil, not from all over the world. So, I think in your title, introduction, results and discussion, you should have mentioned this limitation by saying that the detection method is for the identification of coffee defects from Brazil.

Response: We understand. We have added information on this point in the objective (in the last paragraph of the Introduction) and in the Conclusion (first sentence). We believe it has become clearer.

  1. Preprocessing spectra is one of the important steps when you would like to build a prediction model. Please give an explanation in the methodology of what kind of normalization method you used or any other preprocessing method you have applied in your spectra before analyzing with PLS

Response: We agree. We have included a discussion regarding the preprocessing method in the results and discussion section (Paragraph of line 237). Paragraph of line 209 presents more information on preprocessing method.

Round 2

Reviewer 3 Report

The research paper is interesting.

But I think there are a lot of similar papers to yours.

I do not see anything new that is offered by the authors.

Good luck.

Author Response

We, the authors, disagree with the Reviewer's point of view. No report in the literature performed the analysis using the infrared spectrophotometry technique with photoacoustic detection for this type of sample. Both the analyzed material and the application of the analytical technique for this type of coffee sample are unprecedented. This manuscript highlights the coffee samples, which come from the harvest itself, precisely as the producers use to compose blends. The analyzed samples contain healthy beans (Coffea arabica) mixed with coffee defects such as green, black, and broken beans, and foreign materials such as sticks, husks, and boulders. Infrared-photoacoustic spectroscopy and partial least squares were evaluated as an alternative to the conventional method to measure coffee defect proportion in roasted and ground samples. Multiproduct multivariate calibration models were obtained from spectra samples. All defects were predicted with adequate correlation coefficients. We concluded that the new method is efficient, fast, and suitable for in-line uses for quality control in industrial coffee processing. Thus, this manuscript presents new data on coffee quality monitoring.